# TPA-GEN: A MULTI-MODAL DATA GENERATION METHOD FOR TEXT AND PHYSICS-BASED ANIMATION

## ABSTRACT

Powered by an enormous amount of paired data from the vision and language domains, Vision-Language (V&L) Multi-Modality (MM) research has achieved remarkable results in both text-driven generation and understanding. However, constrained by the data, the learned MM knowledge space predominantly represents the alignments between text and appearances or shapes, lacking further understanding of the underlying dynamics. In this paper, we aim to expand the Multi-Modality (MM) knowledge space by bridging the gap between text, vision, and real-world physical dynamics from a data-centric perspective, enabling MM models to better estimate these dynamics. We propose an automatic pipeline to generate Text-to-Video/Simulation (T2V/S) data. Each generated scenario comprises a high-resolution 3D physical simulation and a textual description of the physical phenomena. To simulate a diverse set of real-world dynamic phenomena—such as elastic deformations, material fractures, collisions, and turbulence—as faithfully as possible, we take advantage of state-of-the-art physical simulation methods: (i) Incremental Potential Contact (IPC) and (ii) Material Point Method (MPM). Additionally, high-quality, multi-view rendering is integrated into the pipeline. We envision our work as the first step towards fully automatic Text-to-Simulation (T2S), potentially shifting the paradigm towards understanding world dynamics.

## 1 INTRODUCTION

In the past few years, we have witnessed the blooming of the Vision-Language (V&L) Multi-Modality (MM) community in solving diverse tasks Lu et al. (2019); Li et al. (2019); Chen et al. (2020); Ramesh et al. (2021); Radford et al. (2021); Zhang et al. (2021); Alayrac et al. (2022); Gao et al. (2022); Kamath et al. (2021); Ramesh et al. (2022). Particularly, V&L models have achieved remarkable performances on various conventional V&L tasks thanks to the availability of an enormous amount of V&L data Schuhmann et al. (2022) and the rapidly developing Large-scale Language Model (LLM) Vaswani et al. (2017); Devlin et al. (2018); Radford et al. (2019); Brown et al. (2020). On the other hand, MM V&L generative tasks are unprecedentedly popular thanks to the advances in Vision-Language (V&L) domain. Text-to-Image (T2I) generation Ho et al. (2020); Ramesh et al. (2021); Rombach et al. (2022); Saharia et al. (2022); Chang et al. (2023) can already produce commercial quality images from free-form text, and meanwhile, Text-to-Video (T2V) Singer et al. (2022); Ho et al. (2022); Khachatryan et al. (2023) and Text-to-3D (T2-3D) Jain et al. (2022); Poole et al. (2022); Jun & Nichol (2023a) are also gaining more attentions.

There are three key factors that jointly contribute to the success of the recent Vision-Language (V&L) research: (i) self-supervised learning techniques and self-attention/cross-attention deep learning architectures are fully explored Vaswani et al. (2017); Devlin et al. (2018); Radford et al. (2019); Lewis et al. (2019); Brown et al. (2020); (ii) Vision-Language (V&L) generative models such as Denoising Diffusion Models (DDM) and Vector-Quantized (VQ) transformer decoder, are well studied Goodfellow et al. (2020); Zhu et al. (2017); Ho et al. (2020); Rombach et al. (2022); Chang et al. (2022; 2023); (iii) Most importantly, a large volume of paired V&L data, *e.g.* Lin et al. (2014); Ordonez et al. (2011); Sharma et al. (2018); Changpinyo et al. (2021); Schuhmann et al. (2022), are available on the Internet, enabling (i) and (ii) to capture the alignments between visual appearance/shape and language tokens' representations.

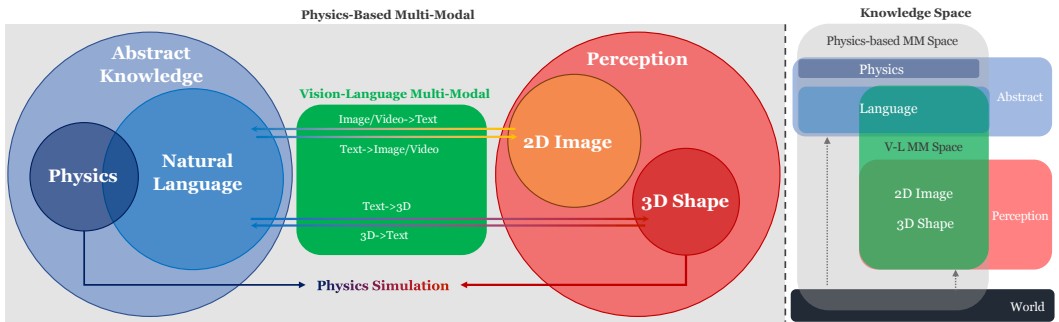

Figure 1: (a) Task space of Physics-based multi-modality. (b) Knowledge space of Physics-based multi-modality.

However, things are not as rosy as they seem. First, as shown in Figure 1(a), current MM V&L paradigm only models the alignments between visual characteristics and corresponding text descriptions. Visual information, such as appearances and shapes, is a projection of the world dynamics to the perception space (see Figure 1(b)). This projection loses a lot of physical information, and the learned V&L representation may not reflect the real world dynamics correctly. It will lead to distortions in both generation and understanding. For instance, the generated video may not be physically realistic. Second, from a statistical perspective, although there is a significant amount of V&L data, it is still far away from fully covering the entire data distribution. In other words, the learned V&L representation cannot cover wide enough real-world phenomena in the spectrum. It will lead to the lack of generalizability and compositionality in text-driven V&L generation. Third, video-text paired data is more domain-specific comparing to image-text paired data. In fact, T2V training oftentimes mixes T2I data. Lastly, the quality of publicly accessible V&L data is varied. A large portion of the data is not usable. Meanwhile, the cost of high quality labeling is too expensive.

As illustrated in Figure 1, our solution is to expand the current V&L MM knowledge space to physics-based MM space where models can directly learn the alignments across text, perception, and physics. In this knowledge space, a Multi-Modality (MM) model can better estimate the world dynamics. In this paper, we propose an automatic data generation pipeline to be the first towards this goal. In each run, our proposed pipeline generates a high-resolution, physically realistic animation with descriptive texts. To cover a wide enough range of physical phenomena, we take the advantages of (i) Incremental Potential Contact (IPC)Li et al. (2020), a robust solid simulation framework that can accurately resolve the intricate contact dynamics for both rigid and deformable objects with guaranteed intersection-free results; (ii) Material Point Method (MPM)Stomakhin et al. (2013); Sulsky et al. (1995), a multi-physics simulation framework that is capable of simulating versatile solids, fluids, granular materials, and multi-physics phenomena. Our pipeline covers various real-world dynamics, such as **deformations**, **fractures**, **collisions**, **turbulence**, *etc*. With commercial-level rendering tools, we also produce high-resolution multi-view videos. To summarize, our automatic data generation pipeline has two major contributions:

- It generates high-quality physically-realistic 3D animations along with sentences describing the physical phenomena, including a wide spectrum of commonly seen real-world dynamics.

- With the generated data, we can expand the current vision-language multi-modal knowledge space to physics-based multi-modal knowledge space. It could help us to better estimate the real-world dynamics behind the scene.

## 2 RELATED WORK

**Text-to-Image and Text-to-Video Generation**  Reed et al. (2016) is recognized as the pioneer in Text-to-Image (T2I) which extends Generative Adversarial Network (GAN) Goodfellow et al. (2020) to multi-modal generation. Similarly, Zhang et al. (2017); Xu et al. (2018) apply GAN variants and further enhance the quality of the generated images with improved image-text alignments. Other works, such as DALL-E Ramesh et al. (2021), formulate the T2I problem as a sequence-to-sequence transfer, and incorporate both Transformer and VQVAE for solutions. Some follow-up

studies show that the results could be further improved by replacing DALL-E components with other vision language modules, such as the CLIP latent space in DALLE2 Ramesh et al. (2022). Moreover, the recent success of Denoising Diffusion Models (DDM) Ho et al. (2020); Rombach et al. (2022) also improves the generation quality with cascading up-sampling diffusion decoder. In Text-to-Video (T2V), most previous works Pan et al. (2017); Li et al. (2018) produce relatively low-resolution videos in simplified domains. Latest research Wu et al. (2021); Hong et al. (2022b); Singer et al. (2022); Ho et al. (2022); Khachatryan et al. (2023) extends the T2I framework to T2V by improving modules in diffusion-based T2I framework, adding additional attention modules, and making use of both image-text and video-text data.

**Text-to-3D, Text-to-Animation Generation and 3D-Text Retrieval**  As extensions of T2I, Dream-Fusion Poole et al. (2022) and Michel et al. (2022) synthesize 3D meshes from texts. Moreover, DreamField Jain et al. (2022) generates radiance field with NeRF. Latest work such as Shap-E Jun & Nichol (2023b) predicts latent parameters for 3D texture and radiance field. Chen et al. (2022) uses texts to control lighting conditions in rendering. Besides, several works use CLIP to enable text-to-3D representations. For example, Khalid et al. (2022) generates mesh and texture in CLIP space; Wang et al. (2022) incorporates CLIP with NeRF, enabling simple text-editable 3D object manipulation; Tevet et al. (2022) generates human motion from text. Hong et al. (2022a) further applies text-to-3D generation to Avatar.

**Vision-Language Datasets**  Microsoft COCOLin et al. (2014), Google Conceptual Captions Sharma et al. (2018); Changpinyo et al. (2021), WIT Srinivasan et al. (2021), and VisualGenome Krishna et al. (2017) *etc*. are most popular fine-labeled image-based V&L datasets. CLEVRJohnson et al. (2017) is one of the iconic synthetic V&L datasets. Besides, billions of image-text pairs have been collected from the internet, such as SBU and LAION 5B Ordonez et al. (2011); Schuhmann et al. (2022). These image-text pair datasets significantly contribute to the success of recent T2I generative models. On the other hand, however, there are less video-text data available, especially fine-annotated video-text datasets. Existing work includes HowTo100M Miech et al. (2019), which mainly focuses on instructional descriptions, and WebViD Bain et al. (2021), which contains high-quality daily activity video clips. Additionally, MSRVTT Xu et al. (2016), MSVD Chen & Dolan (2011), DiDeMo Hendricks et al. (2018), and ActivityNet Caba Heilbron et al. (2015) are commonly used, especially for video-language pre-training. Most of them only contain daily human activity without physical world dynamics.

**Vision-based Physical Reasoning Benchmark**  There is a stream work focuses on vision-based physical reasoning. Although physics are involved into the visual reasoning formulation, they are fundamentally different from our work. CLVERER Yi et al. (2019) and CRAFT Ates et al. (2020) focus on causal relation of rigid-body interaction with simple object primitives. PHYRE Bakhtin et al. (2019) involves more collision-rich and diverse scenes, making the physical reasoning space closer to reality. Our work uses much more advanced physical simulators and includes a much wider range of phenomena. And it is not limited by a simple set of primitive. In fact, we can leverage a much wider range of 3D shapes, making it much more realistic.

## 3 AUTOMATIC TPA GENERATION

As demonstrated in Figure 2, our work employs an attributed stochastic grammar to represent the unified knowledge scenario space that can be instantiated to concrete representations in any modality. Specifically, this tree-structured representation uses *node*s to represent the object-of-interests, and environmental and rendering setups, with different collections of *attribute*s attached to each *node* to represent corresponding properties, as explained in §3.1. The dynamic behaviors of multiple objects are characterized using dynamic models (§3.2) that constrain object velocity properties, as well as multi-object motion and positional relationships. By utilizing constrained sampling, we can obtain a parse tree that represents the initial states and motion characteristics of a concrete scenario, which can then be translated into a physical simulation, rendered videos, and descriptive captions.

The procedure of parse tree sampling is summarized below and elaborated in §3.3. First, the parse tree structure is sampled from the stochastic grammar to decide the content of a scenario. This process will determine the number of simulated objects and collision objects, as well as environmental and rendering setups. Following that, node-related *attribute*s in each hierarchical level will be determined.

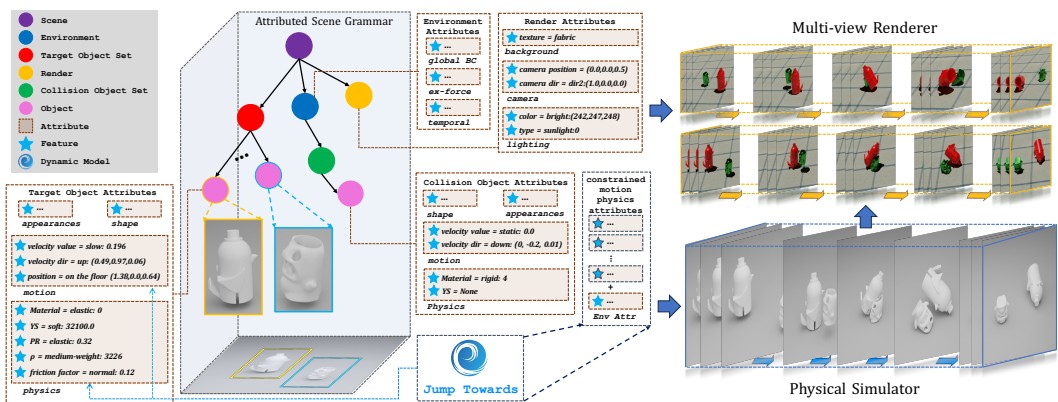

Figure 2: Attributed Scene Grammar that defines the unified knowledge scenario space and its instantiation to 3D-physics and 2D-vision domain.

Afterward, a dynamic model is chosen at random based on the number of objects in the scene, and relation and motion constraints are applied to the *attribute*s of selected objects accordingly.

After settling all *attribute*s and the sampling process finished, we dump this data instance to a JSON file and import it to the IPC or MPM simulator based on the user's choice. IPC can be used to simulate elastic or rigid objects with an accurate friction handler, whereas MPM is adept at handling a variety of materials, including elastic, plastic solids, granular materials, and fluids. The simulator will produce 3D scene representations at multiple discretized time steps. With these data, we can further generate photorealistic videos using an automatic rendering algorithm based on the rendering configurations in the parse tree. More details of the process are introduced in §3.4.

In addition, a stochastic language grammar is constructed from the scenario representation, concurrently with the simulation process (§3.5). A collection of sentences that characterize the animation using randomly selected descriptors is produced as described in §3.6. These sentences are subsequently rewritten using ChatGPT interfaces. In the subsequent subsections, additional modeling and sampling details are introduced.

## 3.1 ATTRIBUTED SCENE GRAMMAR

As previously introduced, we use an attributed stochastic grammar to represent the scenario domain. Specifically, the stochastic grammar is a hierarchical tree composed of the following *node* types: `Scene`, `Target Object Set`, `Collision Object Set`, `Environment`, `Render`, and `Object`. Here, a `Scene` *node* is the root *node* containing three *node*s, `Target Object Set`, `Environment`, and `Render`. Furthermore, `Target Object Set` and `Collision Object Set` are non-terminal *set node*s that can contain an arbitrary number of non-terminal *set node*s of the same type or a random number of `Object` *nodes* that are leaf nodes of the grammar. `Environment` *nodes* are also non-terminal that contain `Collision Object Set` *node*s. Each *node*, according to its categorization, has a particular set of `attribute`s.

`Object` **Nodes.** `Object` nodes belonging to the same *set* may have special semantic relationships, while `Object`s in a specified `Collision Object Set` may constrain the motion and position of `Object`s in a given `Target Object Set`. Moreover, each `Object` *node* contains multiple categories of *attribute*s, including `object-render`, `shape`, `motion`, and `physics`. The *attribute*s are used to specify the corresponding characteristics of the object. Each of them contains several concrete dependent or independent *feature*s that can be directly mapped to a semantic label and a range of quantitative values. For example, `physics` *attribute* consists of three independent *feature*s (material type, friction coefficient, and material density) and two dependent *feature*s (Young's Modulus and Poisson Ratio). The values of the dependent *feature*s rely on the sampling results of both the other independent *feature*s and their own label. In the `physics`, Young's Modulus, for instance, determines the material's resistance to elastic deformation under loads and is therefore dependent on both material types (whether the object is fluids, granular, soft, or rigid solids) and the sampling results of its own label (whether the object is relatively softer or harder).

`Environment` **Nodes.** `Environment` contains *attribute*s to control general scenario configurations such as boundaries, external forces, and temporal discretizations. The `boundary` *attribute* has a `BC` *features* for controlling boundary shape, type, and friction settings, a `Force` *feature* for determining the external force, and a `Time` *feature* for specifying temporal step size and the total number of frames.

`Render` **Nodes.** Expect the `object-render` *attribute* attached to each `Object` to depict the object color and reflective properties, there are additional rendering setups that can reflect the human's visual imagination of a given scenario. We use a terminal *node*, `Render`, to specify those configurations, such as background light, textures, and the position of the camera. All of these setups are, as before, supported by *attribute*s with detailed *features*.

### 3.2 DYNAMIC MODEL

We propose dynamic models to characterize and constrain object motions and relationships in addition to the tree-structure grammar. Each dynamic model can be mapped to a **verb** that semantically describes the velocity feature and interactions between **subjective** and **objective** objects. It may also include directional descriptors such as `from` and `to` to further guide the objects' moving characteristics and initial position properties.

Currently, our data generative model supports the following dynamic models: `JUMP`, `DROP`, `THROW`, `PUSH` and `STRIKE`. The first three models, which are referred to by intransitive verbs, are capable of influencing the behavior of one or two objects. If a single object is sampled, these models will either constrain the initial position of the object to be on the ground or in the air by confining the corresponding position *feature* in the `motion` *attribute*. In addition, they will assist the object in choosing an appropriate velocity scale and movement direction. If two `Objects` are sampled in the corresponding *set nodes*, however, a directional descriptor will be sampled to customize the relationship between these two objects. One of them is selected at random to serve as the subjective object, while the remainder severs as the objective. Their relationship will be constrained by the selected directional descriptor. `From`, for instance, indicates that the initial position of the subjective object is close to the objective object and that it is moving in the opposite direction; whereas `to` means that the subjective object starts from a relatively distant location and moves toward the objective.

The `PUSH` and `STRIKE` models are slightly distinct due to the transitive nature of these verbs. This suggests that they inherently associate an objective object with the subject described. The semantic meaning of the verbs also constrains the initial positions and motion directions of the involved objects. If directional descriptors are also sampled within the transitive dynamic models, an extra object will be introduced with additional constraints. As a case study, we can sample a model that reads "*sub* `PUSH` *obj_1* to *obj_2*". This can be interpreted as the *sub* moving toward *obj_1* with velocities pointing to *obj_2* and trying to push *obj_1* in the direction of *obj_2*.

In practice, our model can be expanded by integrating additional dynamic models with minimal design and implementation effort. In human languages, all verb semantics are subjectively defined, necessitating the manual design of object relations and feature constraints. As we offer a variety of constraint/relation/feature-related abstract interfaces for defining, validating, and applying each user-defined constraint, it is straightforward to convert the design of a constraint set into a dynamic model class in our codebase.

### 3.3 SCENARIO SAMPLE PROCESS

In order to instantiate a concrete scenario from the stochastic grammar, we need to sample 1) a parse tree structure that defines the content and characteristics of a scene, 2) the concrete qualitative labels and quantitative values of *features* in corresponding *attribute*s, 3) a dynamic model with a specified verb and optional directional descriptor that constrains the dynamic behavior of several objects. This section examines these three phases in detail.

**Sample Parse Tree Structure.** The structure sampling procedure begins at the root *node* and progresses downward until it reaches the terminal *node*s. Different *node*s are sampled according to their individual categories in order to specify which children *node*s are essential and which are optional. *Set node*s, for instance, can sample any number of children *node*s within a permitted children number range. In contrast, an `Environment` *node* must contain at least one `Collision`

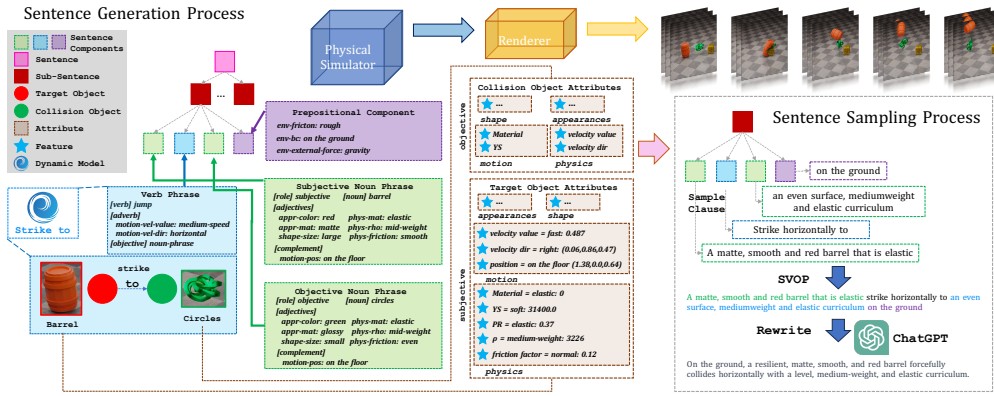

Figure 3: Sentence generation process.

`Object Set` node. Following this phase, the total number of object-of-interest and collision objects will be determined. This information will further narrow our selection of dynamic models.

**Sample *Feature*s.** As specified in §3.1, each *node* in the parse tree comprises a number of *attribute*s. Additionally, an *attribute* contains multiple *feature*s to determine particular semantic properties. In this stage, the goal is to first sample a qualitative label for each *feature*, then sample the corresponding absolute values that define certain physical or visual properties.

To attain this objective, we begin by randomly selecting *feature* labels in each tree hierarchy using in a top-down sampling manner. The independent *feature*s are sampled from the candidate pool, while the dependent *feature*s are sampled subsequently to ensure semantic consistency in describing the scenario. Following the selection of all *feature* labels, concrete *feature* values are sampled from the predefined quantitative ranges of each label.

Top-down sampling provides only an initial selection of *feature*s; the final labels and values are further determined by the choice of dynamic models (next paragraph). With the selected model, all concerned *feature*s will undergo a bottom-up refinement. We first evaluate whether the values of specified *feature*s satisfy the dynamic model's constraints. If not, the out-of-range projection of the values will be removed in order to enforce the constraints. Once all constraints are met, we reselect the *feature* labels based on the modified values.

**Sample Dynamic Models.** As indicated in §3.2, diverse dynamic models are capable of guiding a variety of object counts. In this step, the choice of the dynamic model is based on the total number of objects (including object-of-interests and collision objects). The total number of objects must be sufficient to accommodate the motion and relationship depicted by the current model. Afterward, we randomly select objects from the parse tree to serve as the subjective or objective(s) of the respective motion. Note that the subject can only refer to the object-of-interest, whereas the objective can be of any types. If additional free objects or collision items remain, they will be regarded as noise unrelated to the current scenario data point and will not be included in the language model (§3.5).

## 3.4    SIMULATION AND RENDERING

After determining the scene parse tree structure with appropriate *feature* labels and values, the data point is transferred into a `JSON` format. This output `JSON` file is then sent to an Incremental Potential Contact (IPC) Li et al. (2020) or an Material Point Method (MPM) Qiu et al. (2022) simulator based on the sampled object materials or the user's preferences. According to the `JSON`, physical simulators initially load object shapes (§4) and assign both object- and environment-related parameters from corresponding *feature* values. Then, the object motion and material behaviors such as deformation and fractures are simulated until the maximum frame number is reached.

A renderer then collects the 3D output results at various time steps to generate high-fidelity rendering results. In addition, rendering configurations such as background texture, object colors, reflective materials, background light, and camera position are also loaded from the sampled `JSON` file.

Blender Blender (2018), which is open-sourced and supports fully Python-scriptable rendering operations, is used to accomplish automatic rendering.

In addition to the technologies employed by the proposed pipeline, other publicly available simulators (*e.g.*, NVIDIA's FleX Macklin et al. (2014)) and rendering engines can also be utilized with corresponding `JSON` file parser.

## 3.5 LANGUAGE GENERATION MODEL

On the basis of the sampled scenario parse tree, a hierarchical tree-structured language model is constructed (Figure 3). In this model, the root node, which represents a *sentence* structure, is decomposable into multiple *sub-sentence* nodes. Each *sub-sentence* consists of three nodes representing typical linguistic components: a *prepositional* node, a *noun phrase* node, and a *verb phrase* node. In this case, the *prepositional* node collects feature labels associated with environmental and rendering configurations in the parse tree, thereby describing the global scenario characteristics. If the *noun phrase* node is placed under the root node, it is considered the subjective object in the scene; otherwise, if belonging to a *verb phrase*, it is regarded as the objective object. Detail-wise, a *noun phrase* contains a *noun* and its descriptors which are summarized from the corresponding `Object` *node*. And *verb phrase*, on the other hand, has a *verb* with dynamic descriptors and multiple *noun phrase* children nodes performing the objective roles.

When constructing the language model, we begin by examining the type of dynamic model and mapping it to a verb in the *verb phrase*; the object relationships in the dynamic model determine which `Object` falls to the *subjective phrase* and which refers to the *objectives*. Then, if the *features* in the parse tree and dynamic model merit being stated in sentences, they are assigned to language components. Specifically, the shape *feature* of an `Object` is captured by certain *noun phrases*, and the tags of the corresponding object mesh are retrieved and sampled as the noun. The other physical and rendering features are attached as adjective descriptors to the *noun phrase*. Certain specialized *feature*s, such as Young's Modulus, are too specific to be included in common language and are therefore neglected. Additionally, the verb phrase collects the subjective `Object`'s velocity-related features as auxiliaries. And finally, *feature*s associated with the `Environment` and `Render` *node*s are inserted into the *prepositional* node.

## 3.6 GENERATING RANDOM SENTENCES

In addition to the aforementioned structured language model, the next stage is to create concrete sentences that describe the scenario. The entire sentence consists of sub-sentences joined by conjunctions such as ",", ";" or "and". Additionally, each sub-sentence is composed of a subjective phrase, a verb phrase, and prepositional components. To obtain each finalized sub-sentence, we break down the problem into multiple steps, which are described in the paragraphs that follow.

**Sample Sentence Structure.** The components of the sub-sentences can appear in a variety of arrangements to create diverse sentence structures. The objective of this phase is to determine this order. We offer several common sentence structures as candidates. For example, SVOP (Subject-Verb-Object-Preposition) is the most common English sentence structure, whereas OVS is an example of passive voice.

**Sample Components.** In this step, each sentence component is sampled independently into concrete clauses and concatenated in the order specified by the predefined sentence structure. Using the language model constructed in §3.5, the corresponding clauses can be formed. As stated previously, each language component node contains all the accumulated feature labels from the parse tree. However, including all potential descriptive terms in our everyday language is unnecessary and cumbersome. Therefore, we arbitrarily select a number of descriptors that appear either before or after the noun/verb in corresponding *phrases*.

Specifically, for *noun-phrases*, descriptors displayed before nouns are adjectives joined by conjunctions, whereas descriptors presented after nouns can be formulated as subclauses introduced by "that" or "which". The descriptor number can also be zero, indicating that the noun contains no description portion. As an instance, we can sample two adjectives and three clause-descriptors from the subjective *noun phrase* to construct a noun phrase clause such as "a blue and matte cube that is small, elastic, and rough". The phrases "blue and matte", "cube", and "small, elastic and rough" denote the adjective-, noun-, and clauses-portions, respectively.

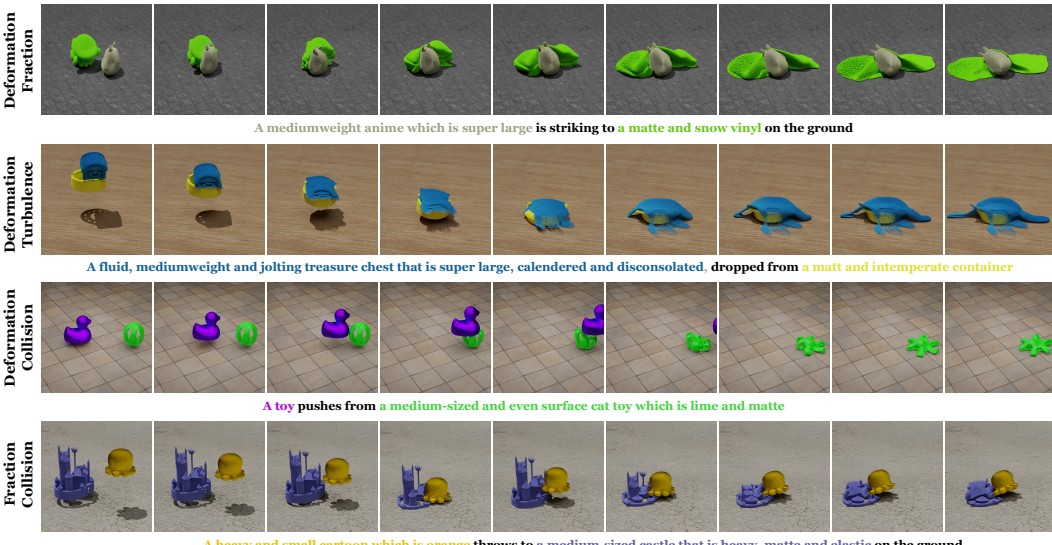

Figure 4: Examples of generated animations.

Similar strategies are used to manage descriptors in the verb-phrase case. To enhance sentence diversity, we sample the verb tense further when generating the verb phrase clause. The objective object of the verb is sampled as another noun phrase. As for the prepositional portion, a random sample of conjunction-coupled labels is selected.

**Sentence Diversity.**   To maximize the diversity of the sampled sentences, we sample them multiple times so that various types and quantities of object and motion descriptors are chosen. Then, ChatGPT interfaces are called to rewrite each sentence, with suitable prompts. In this sense, both sentence structure and word synonyms are interchangeable.

## 4   3D SHAPE COLLECTING AND PROCESSING

Apart from the generation procedure described in the §3, additional support is required to complete the pipeline. That is, to collect, process, and utilize 3D shapes with noun labels indicating what the shape represents. This section describes three methods for achieving this objective, along with their advantages and disadvantages.

**3D Object From Existing Dataset.**   Existing 3D object datasets, such as Thingi10KZhou & Jacobson (2016), can be used as the prospective shape pool. Preprocessing is required to meet the input format specifications of Incremental Potential Contact (IPC) and Material Point Method (MPM). Particularly, IPC accepts a tetrahedral geometry (4 vertices per face, .ply format) whereas MPM accepts a 3D volumetric signed distance field (.vdb format).

Thanks to the contributions of Zhou & Jacobson (2016), we can readily collect a large number of 3D forms. Nevertheless, noun sampling with Thingi10K is a laborious procedure. Specifically, we use properties like titles and tags affixed to all the shapes as the object noun vocabulary. However, the descriptiveness and quality of these terms are less reliable (containing adjectives like "funny", "movable" and over-broad concepts like "3D", or "art").

**Text-based 3D Shape Generation.**   Another alternative is to generate 3D shapes from text labels. We first ask ChatGPT to generate a certain number of nouns of a specific type, and then feed these words into a text-to-3D model to generate 3D shapes. In our experiments, we use Shap-EJun & Nichol (2023b), while other similar generators could also be viable alternatives.

This procedure mitigates the disadvantage described in the preceding paragraph. However, new difficulties arise. First, the majority of models for generating 3D shapes require carefully adjusted parameters for desired results, which doesn't fit our requirement for automatic mass production. Therefore, the generated 3D meshes are of unreliable quality. To filter out invalid meshes that are

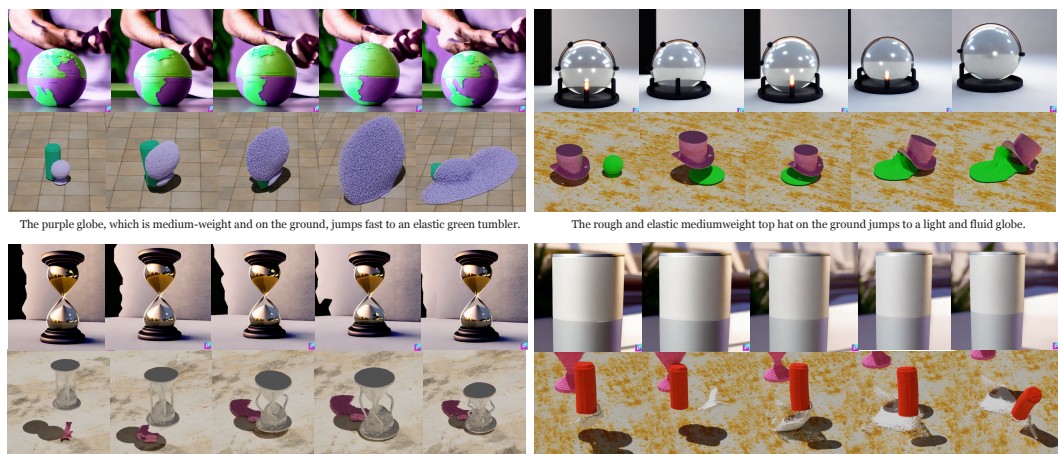



The purple globe, which is medium-weight and on the ground, jumps fast to an elastic green tumbler.   The rough and elastic mediumweight top hat on the ground jumps to a light and fluid globe.

A medium-weight hourglass fell from the sky and landed on an anvil.   The smooth and matte tumbler quickly struck the light, white, sandy quill in the air



Figure 5: Side-by-side comparison of generated animation and zeroshot Text-to-Video (T2V).

noisy, incomplete, or contain too many holes, we designate all generated shapes with a command-line labeling tool.

**Text-based 3D Shape Retrieval**   Utilizing an existing 3D shape retrieval model (*e.g*. Liu et al. (2023)) to search and extract existing shapes from a pre-sampled noun list is an alternative solution. The quality of the retrieved meshes is trustworthy, and the noun we employed for retrieval can serve as the noun descriptor in our sample sentences. We are eager to incorporate this work into our pipeline once their code has been published.

## 5   QUALITATIVE COMPARISON OF T2V GENERATION

We list some representative rendering results with descriptive captions in Figure 4. More demos can be found in the supplementary materials. Due to the absences of benchmark methods, we set up an qualitative comparison between generated animation and zero-shot text-to-video generation results. Although it may not be a fair comparison, it still conveys our the main idea of the proposed method. As shown in Figure 5, the zero-shot Text-to-Video (T2V) Khachatryan et al. (2023) generated results shows very limited dynamic interactions between the involved objects and the world. Oftentimes the video shows no dynamics but simply slight viewpoint shifts. However, our generated animations show vivid physical dynamic interaction across the scene. As mentioned in §1, such a difference is cause by absences of modeling physical knowledge in the Multi-Modality (MM) knowledge space.

**Why no quantitative evaluation?**   The reasons are two folds: (i) There are no feasible methods or formulations that can fully leverage our data. Specifically, none of the generative model or understanding model consider physical constraints and elasto-plastic material descriptions. (ii) Current automatic metrics are not reliable. In computer vision, metrics such as FID, FVD and average CLIP score are often-time used to measure the similarity between image/videos. However, none of them take physical fidelity into consideration. As shown in Figure 5, although similar objects appears in both results, the physical fidelity are radically different.

## 6   CONCLUSION

In this study, we have introduced an innovative approach for automatically generating physics-based animations with textual descriptions. Our method has been extensively analyzed, and we have presented both qualitative results of our data generation method and comprehensive experiments that highlight the importance of such physically realistic datasets to the multi-modal generation research community. We believe that the addition of these resources could substantially contribute to the expansion of the current vision-language multi-modal knowledge space, facilitating improved understandings and estimations of real-world dynamics.

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
