# TPA-GEN: A MULTI-MODAL DATA GENERATION METHOD FOR TEXT AND PHYSICS-BASED ANIMATION

## A  SCENE GRAMMAR PRODUCTIONS

We use an attributed stochastic grammar as a hierarchical and structured representation that determines the scenario's content with initial physical parameters and appearance settings. The grammar is decomposed into multiple levels of components which are sampled according to the production rules defined in Table 1. The tree structure itself describes the scenario's content, while the related `attributes`, which contain numerous `features`, specialize the content's characteristics. Table 2 presents a list of the `attributes` and `features` designed for each *node*.

| Label | Production Rules |
|---|---|
| Scene | Scene → TarObjSet ⊕ Environment ⊕ Render |
| Component-0 | TarObjSet → TargetObj$^+$ ⊙ TarObjSet$^*$ |
| | Environment → ColObjSet |
| Component-$\star$ | ColObjSet → CollisionObj$^*$ ⊙ ColObjSet$^*$ |
| | TarObjSet$^*$ → TargetObj$^*$ ⊙ TarObjSet$^*$ |

Table 1: **Production rule of the scenario stochastic grammar.** Here, `TarObjSet` is short for `Target Object Set` which includes a set of simulated object (`TargetObj`) with potential relationships; `ColObjSet` represents a set of non-movable collision objects (`CollisionObj`) serving as boundary conditions. Moreover, ⊕ represents *and* relation, making the child elements mandatory; while ⊙ refers to *or* relation to connect optional child nodes; $^+$ means one or more and $^*$ means zero or more.

## B  DYNAMIC MODEL AND CONSTRAINTS

### B.1  CONSTRAINTS

In practice, we use the following eight constraints to reveal object relationships and the constraints are checked and applied on selected object `features`. Every constraint consists of a list of operands "$[o_0, o_1, ..., o_N]$" and an ID number list "$[n_0, .., n_M]$" which represents the unalterable criteria operand(s). Here, operand $o_i$ refers to either a constant (value or vector that is always unalterable), or a *node*-`attribute`-`feature` pair in which case the value of the corresponding feature is fetched for computation. Additionally, the criteria operands must by default satisfy the given constraints to avoid ambiguities.

- *less_eq*($[o_0, ..., o_N], [n_0, ..., n_M]$): $o_0 \leq ... \leq o_N$, with $o_{n_0}, ... o_{n_M}$ stay unchanged during resampling.

- *less*($[o_0, ..., o_N], [n_0, ..., n_M]$): $o_0 < ... < o_N$, with $o_{n_0}, ... o_{n_M}$ stay unchanged during resampling.

- *larger_eq*($[o_0, ..., o_N], [n_0, ..., n_M]$): $o_0 \geq ... \geq o_N$, with $o_{n_0}, ... o_{n_M}$ stay unchanged during resampling.

- *larger*($[o_0, ..., o_N], [n_0, ..., n_M]$): $o_0 > ... > o_N$, with $o_{n_0}, ... o_{n_M}$ stay unchanged during resampling.

- *eq*($[o_0, ..., o_N], [n_0, ..., n_M]$): $o_0 = ... = o_N$, with $o_{n_0}, ... o_{n_M}$ stays unchanged during resampling.

- *same_dir*($[o_0, ..., o_N], [n_0]$): $o_i$ must be vectors, and for $\forall i \in [0, ..., N]$ the angle between $o_i$ and $o_{n_0}$ is zero. Note that only one criterion operand is allowed to be present in this constraint.
- *oppo_dir*($[o_0, ..., o_N], [n_0]$): $o_i$ must be vectors, and for $\forall i \in [0, ..., N]$ the angle between $o_i$ and $o_{n_0}$ is $180°$. Note that only one criterion operand is allowed to be present in this constraint.
- *similar_dir*($[o_0, ..., o_N], [n_0, ..., n_M], \theta$): $o_i$ must be vectors, and for $\forall i \in [0, ..., N]$, $\forall j \in [0, ..., M]$ the angle between $o_i$ and $o_j$ is less or equal to $\theta$. Here, $o_{n_0}, ... o_{n_M}$ stays unchanged during resampling.

During sampling process, we validate the defined constraints after random sampling all `features`. The non-criteria operands that violate the constraints will be resampled to guarantee the correctness of the relation. If the criteria operands themselves violate the constraint, the resample process will be terminated and errors will be reported.

### B.2 DYNAMIC MODEL

As introduced in the main paper, we have the following dynamic models: `JUMP`, `DROP`, `THROW`, `PUSH` and `STRIKE`. We summarize the basic constraints required for each intransitive dynamic model in Table 3 and transitive dynamical verbs in Table 4. In additional to these dynamic models, one can easily define and generate other dynamic models into our codebase with predefined interfaces for constraints. Some example defintions can be found in Table 5.

In all the above mentioned tables, "DM" refers to Dynamic Model, and the "sub" is used to denote the subjective object on which the verb in dynamic model focuses. And as shown in the last two columns of the table, objective objects (represented by "obj") are introduced with `from` and `to` directional relations.

## C DATASHEET FOR DATASET

We answer the questions that applied to our work:

### C.1 MOTIVATION

- **For what purpose was the dataset created?** We propose a method to generate Text and Physics-based Animation (TPA) for multi-modal model training. The goal is to expand the current problem domain of multi-modal learning from image-text understanding to vision-world dynamics understanding. We believe that this is the one of the beginning steps to enable multi-modal model's capability to understand our world from a model fundamental perspective.
- **Who created the dataset and on behalf of which entity?** The authors are from three different institutes. Yuxing Qiu, Minchen Li and Chenfanfu Jiang are from UCLA Multi-Physics Lagrangian-Eulerian Simulation Lab (MultiPLES). Feng Gao, was a Ph.D. student at UCLA when the project was initiated. Yin Yang is from University of Utah, Utah graphics lab. Govind Thattai joins the project as a individual contributor.
- This work is fully funded by UCLA computer science department and UCLA MultiPLES lab.

### C.2 COMPOSITION

- **What do the instances that comprise the dataset represent?** Each generated instance consists of the following elements: a video of the rendered physics-based animation, a set of text descriptions of the animation, a 3D material points (position) of the object ID at frame ID.
- **How many instances are there in total?** As mentioned in the main paper, we propose a method to generate TPA data. In theory, one can generate as much as possible of TPA data with sufficient objects, types of materials and pre-defined motion types. To better illustrate the details of the data, we provide a sample set of TPA data which contains 500 instances.

- **Does the dataset contain all possible instances or is it a sample of instances from a larger set?** No.

- **Are there recommended data splits?** We don't specify any splits. One can configure the split accordingly.

## C.3 COLLECTION PROCESS

As we mentioned in the main paper, we propose a method to generate text and physics-based animation data. Since the data is synthetic, we don't require any human annotators to involve. Besides, in order to improve the quality of the generated texts, we take advantages of the most popular large language model, ChatGPT, to rewrite the generated description of the animation. Beside, we also use ChatGPT to help proposing label names of the 3D objects in a TPA instance. The 3D objects are generated with a SoTA text-3D generation model. Specifically we use the checkpoint of GPT-3.5-turbo-0301 as the backbone model.

## C.4 PRE-PROCESSING/CLEANING/LABELING

- **3D object representation transformation**: The 3D objects are spatially discretized to material points at each time step. The positions of the material points representing each object (including object-of-interests and collision object)d are stored in separate PLY files.

- **Description rewriting:** We use ChatGPT to help cleaning the generated text description by rewriting the sentences without changing the meaning of it.

## C.5 DISTRIBUTION

- **How will the dataset be distributed?** The implementation of the propose TPA data generation method contains three parts: 1, a scene sampling process code; 2, a binary execution of a physics-based simulation engine complied from a set of source code. The source code of this simulation engine is based on a open-source version of Material Point Method (MPM); 3, data cleaning codes. Above implementations will be made public on GitHub.

- **When will the dataset be distributed?** The sample dataset and the implementation code will be distributed after the NeurIPS dataset track review process.

## C.6 MAINTENANCE

- **Who will be supporting/hosting/maintaining the dataset?** UCLA MultiPLES lab will be the host of the proposed dataset generation method.

- **How can the owner of the dataset be contacted?** Yuxing Qiu (yuxqiu[AT]g.ucla.edu) and Chenfanfu Jiang (cffjiang[AT]math.ucla.edu) are the main contacts of the proposed data generation method.

- **Will the dataset be updated?** UCLA MultiPLES lab will keep updating the generation method including adding more 3D objects and types of motions into the generation process.

# D DATASET SUBMISSION REQUIREMENTS

## D.1 IMPACT AND CHALLENGES

We expect our proposed method and the data generated by this method can make a board impact to both multi-modal and computer graphics community.

- In terms of multi-modal understand, as discussed in the main paper, we aim to help this community to expand the problem domain from shallow vision-language alignment to deep comprehension of the knowledge space of vision-language-world dynamics. It could boardly impact specific research domain such as Text-to-Video/Simulation (T2V/S), robotics and intuitive physics.

- Our method could also make a large impact in the conventional computer graphics domain by reformulating the 3D animation creation process. Our generation process provides a rule-based generation of 3D animation scenes. On the other hand, once we have a reliable model that can generate 3D animation from human language, it could save huge amount of efforts to create 3D animation from scratch. The traditional pipeline usually requires very experienced artist and computer graphics researchers and engineers to collaborate even for a simple scene. Our method initialize the very first step towards fully automatic generation of 3D physics-based animations from text description.

## D.2 Licensing ans Access

We would like to specify that we intend to utilize the **MIT license** for the method proposed in this paper to generate TPA data. This open-source license grants users the freedom to use, modify, and distribute the dataset while providing clear attribution to the original creators. By choosing the MIT license, I aim to foster collaboration, encourage innovation, and ensure that the generated data the code of the proposed method to generate remains accessible to the wider community for further exploration and development.

## E Data Samples

Figure 1, Figure 2, Figure 3, and Figure 4 demonstrate more sample examples of the proposed generative algorithm. For more demos, please check out this google drive: https://drive.google.com/drive/folders/1IbPJBmPLlzB4DPmXVx1eQhSr42WYjLU_?usp=sharing.
One can check out the demos in the following file hierarchy (DM refers to concrete dynamic model names):

- DM_scene:
    - label_out.json (labels of all nodes, attributes and features)
    - value_out.json (quantitative values of corresponding setups)
    - sentence_original.json (sentence sampled from the proposed language model)
    - sentence_rewrite.json (sentence rewritten by ChatGPT)
    - render (a folder contains rendered results)
        * FID.png (FID refers to frame ID)
        * out.mp4 (video of rendering results)
- DM_3d: (3D object data)
    - _0_0_0_target_OID_FID.ply (OID refers to object ID; 3D material points (position) of object OID at frame FID)
    - _0_0_0_collision_FID.ply (point positions of collision objects)

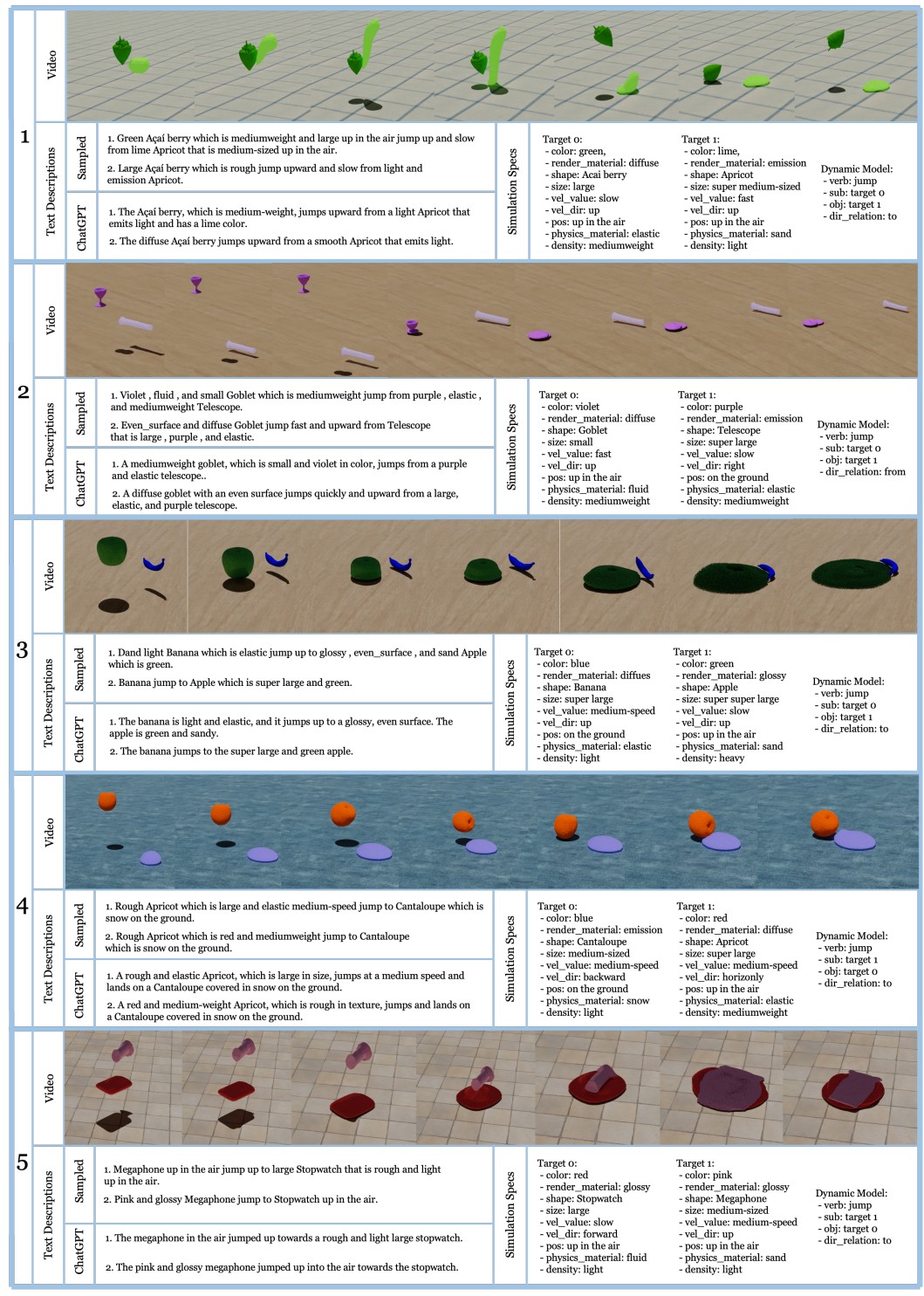

Figure 1: Data samples of **DROP** dynamics.

| *Node* | Attribute | Feature | Label Candidates |
|---|---|---|---|
| Env | Boundary Condition | BOUNDARY | Box, Floor |
| | | TYPE | Sticky, Slip |
| | | FRICTION FACTOR | Smooth, Even Surface, Rough, Extremely rough |
| | External Force | FORCE TYPE | Gravity, Wind |
| | | *Force value* | *Dependent on* FORCE TYPE |
| | Temporal | TOTAL FRAME | Short, Medium, Long |
| Object | Appearance | COLOR | White, Red, Blue, Green, Lime, Orange, Yellow, Pink, Purple ... |
| | | MATERIAL | Glossy, Matte |
| | Shape | SHAPE | Cube, Sphere, Cylinder, Mesh |
| | | SIZE | Small, Medium-sized, Large, Super large |
| | Motion | VELOCITY VALUE | Slow, Medium-speed, Fast |
| | | VELOCITY DIRECTION | Up, Down, Right, Left, Forward, Backward, Horizontal, Vertical |
| | | INITIAL POSITION | On the ground, In the sky |
| | Physics | MATERIAL | Elastic, Rigid, Fluid, Snow, Mud, Sand, Granular |
| | | *Young's Modulus* | *Dependent on* MATERIAL; Soft, Moderate-hardness, Hard, Rigid |
| | | *Poisson Ratio* | *Dependent on* MATERIAL; Elastic, Rigid |
| | | DENSITY | Light, Medium-weight, Heavy |
| | | FRICTION FACTOR | Smooth, Even Surface, Rough, Extremely rough |
| Render | Background | LIGHT | Bright, Dark |
| | | TEXTURE | Preset texture list or "random" |
| | Camera | CAMERA POSITION | Preset camera position |
| | | *Viewpoint* | *Dependent on* CAMERA POSITION |

Table 2: **Attributes with features associated for each scene *node*.** In the table, independent features are highlighted with the SMALL CAPS font style, whereas dependent features are labeled with *italic*. The final column lists examples of candidate labels for each feature. Each label is mapped to a specific value or range of values based on its semantics. Additional labels can be easily appended by providing a mapping between the label name and corresponding value ranges.

| DM | Type | Constraint |
|---|---|---|
| JUMP | Basic | $similar\_dir([[0,1,0],(\text{sub, Motion, VELOCITY DIRECTION})],[0],\theta_0)$
$less\_eq([v_{min},(\text{sub, Motion, VELOCITY VALUE})],[0])$
($v_{min}$ defined by user) |
| | from | $eq([\mathbf{p}^{\text{gt}},(\text{sub, Motion, INITIAL POSITION})],[0])$, with $\mathbf{p}^{\text{gt}}=[p_0^{\text{gt}},p_1^{\text{gt}},p_2^{\text{gt}}]$
Here, $p_i^{\text{gt}}=p_i^{\text{obj}}\pm(s_i^{\text{sub}}+s_i^{\text{obj}}+C)*0.5$ for $i\in[0,2]$ |
| | to | $similar\_dir([\mathbf{d}^{\text{gt}},(\text{sub, Motion, VELOCITY DIRECTION})],[0],\theta_1)$
Here, $\mathbf{d}^{\text{gt}}=(\mathbf{p}^{\text{obj}}-\mathbf{p}^{\text{sub}})+\alpha\cdot[0,1,0]$ ($\alpha$ defined by user) |
| DROP | Basic | $similar\_dir([[0,-1,0],(\text{sub, Motion, VELOCITY DIRECTION})],[0],\theta_0)$
$larger\_eq([v_{\text{small}},(\text{sub, Motion, VELOCITY VALUE})],[0])$
($v_{\text{small}}$ defined by user)
$less\_eq([\mathbf{p}^{\text{gt}},(\text{sub, Motion, INITIAL POSITION})],[0])$,
$\mathbf{p}^{\text{gt}}=[p_0^{\text{gt}},p_1^{\text{gt}},p_2^{\text{gt}}]$;   $p_1^{\text{gt}}$ is user-defined threshold;
$p_0^{\text{gt}}$ and $p_2^{\text{gt}}$ are the global minimum position |
| | from | $eq([\mathbf{p}^{\text{gt}},(\text{obj, Motion, INITIAL POSITION})],[0])$
Here, $p_i^{\text{gt}}=p_i^{\text{sub}}\pm(s_i^{\text{sub}}+s_i^{\text{obj}}+C)*0.5$ for $i\in[0,2]$ |
| | to | $eq([\mathbf{p}^{\text{gt}},(\text{obj, Motion, INITIAL POSITION})],[0])$
Here, $p_i^{\text{gt}}=p_i^{\text{sub}}\pm(s_i^{\text{sub}}+s_i^{\text{obj}}+C)*0.5$ for $i=0,2$; $p_1^{\text{gt}}=s_1^{\text{obj}}+C$ |
| THROW | Basic
(UP) | $similar\_dir([\mathbf{v}_{dir}^{\text{gt}},(\text{sub, Motion, VELOCITY DIRECTION})],[0],\theta_0)$,
$\mathbf{v}_{dir}^{\text{gt}}=[C_0,1,C_1]$ |
| | Basic
(DOWN) | $similar\_dir([\mathbf{v}_{dir}^{\text{gt}},(\text{sub, Motion, VELOCITY DIRECTION})],[0],\theta_0)$,
$\mathbf{v}_{dir}^{\text{gt}}=[C_0,-1,C_1]$
$eq([\mathbf{p}^{\text{gt}},(\text{obj, Motion, INITIAL POSITION})],[0])$.
Here, $p_i^{\text{gt}}=p_i^{\text{sub}}\pm(s_i^{\text{sub}}+s_i^{\text{obj}}+C)*0.5$ |
| | from | $eq([\mathbf{p}^{\text{gt}},(\text{obj, Motion, INITIAL POSITION})],[0])$
Here, $p_i^{\text{gt}}=p_i^{\text{sub}}\pm(s_i^{\text{sub}}+s_i^{\text{obj}}+C)*0.5$ for $i\in[0,2]$ |
| | to | $eq([\mathbf{p}^{\text{gt}},(\text{obj, Motion, INITIAL POSITION})],[0])$
Here, $\mathbf{p}^{\text{gt}}=\mathbf{p}^{\text{sub}}+\mathbf{s}^{\text{sub}}+v_{value}^{\text{sub}}\cdot\mathbf{v}_{dir}^{\text{sub}}\cdot C$ |

Table 3: **Constraints in each single-object dynamic model.** In the table, $\mathbf{s}$, $\mathbf{p}$, $v_{value}$ and $\mathbf{v}_{dir}$ refers to object size, initial position, velocity value and velocity direction, separately; $C\geq 0, C_0\in[-1,1], C_1\in[-1,1], C_{small}\in[0,0.1*s_{max}^{sub}]$ represents random noise, and $\pm$ means +/- are chosen randomly in practice. All the dynamic model has the basic constraints applied to the objects, with randomly sampled `from`, `to`, or NONE relation. Note that `from` and `to` relation will be chosen only when there are enough objects in the scenario. Specially, for THROW model, we will first sample to decide if it is "Throw UP" or "Throw DOWN" before defining the basic constraints.

| DM | Type | Constraint |
|---|---|---|
| PUSH | Basic | $similar\_dir([\mathbf{v}_{dir}^{\text{gt}}, (\text{sub, Motion, VELOCITY DIRECTION})], [0], \theta_0)$
Here $\mathbf{v}_{dir}^{\text{gt}} = \mathbf{p}^{obj} - \mathbf{p}^{\text{gt-sub}}$
$less\_eq([v_{\text{large}}, (\text{sub, Motion, VELOCITY VALUE})], [0])$
($v_{\text{large}}$ defined by user)
$larger\_eq([v_{\text{small}}, (\text{obj, Motion, VELOCITY VALUE})], [0])$
($v_{\text{small}}$ defined by user) |
| | Basic
(Not from) | $eq([\mathbf{p}^{\text{gt-sub}}, (\text{sub, Motion, INITIAL POSITION})], [0])$
Here, $p_i^{\text{gt-sub}} = p_i^{\text{obj}} \pm (s_i^{\text{sub}} + s_i^{\text{obj}} + C) * 0.5$ for $i \in [0, 2]$ |
| | from | $eq([\mathbf{p}^{\text{gt-sub}}, (\text{sub, Motion, INITIAL POSITION})], [0])$
Here, $p_i^{\text{gt-sub}} = p_i^{\text{obj-extra}} \pm (s_i^{\text{sub}} + s_i^{\text{obj-extra}} + C) * 0.5$ for $i \in [0, 2]$
$eq([\mathbf{p}^{\text{gt-obj}}, (\text{obj, Motion, INITIAL POSITION})], [0])$
Here, $\mathbf{p}^{\text{gt-obj}} = \mathbf{p}_i^{\text{gt-sub}} + K * \frac{\mathbf{p}^{\text{gt-sub}} - \mathbf{p}^{\text{obj-extra}}}{||\mathbf{p}^{\text{gt-sub}} - \mathbf{p}^{\text{obj-extra}}||}$
$K = 0.5 \cdot \max_{i=0,1,2}(s_i^{\text{sub}} + s_i^{\text{obj}}) + C$ |
| | to | $eq([\mathbf{p}^{\text{gt-obj-extra}}, (\text{obj-extra, Motion, INITIAL POSITION})], [0])$
Here, $\mathbf{p}^{\text{gt-obj-extra}} = \mathbf{p}^{obj} + K * \mathbf{v}_{dir}^{\text{sub}}$
$K = 0.5 \cdot \max_{i=0,1,2}(s_i^{\text{obj}} + s_i^{\text{obj-extra}}) + C$ |
| STRIKE | to | $\mathbf{p}^{to} = \mathbf{p}^{\text{obj-extra}}$ |
| | Not to | Random sample $\mathbf{p}^{to}$ |
| | Basic | $similar\_dir([\mathbf{p}^{\text{to}} - \mathbf{p}^{\text{sub}}, ($
sub, Motion, VELOCITY DIRECTION$)], [0], \theta_0)$
$similar\_dir([\mathbf{p}^{\text{to}} - \mathbf{p}^{\text{obj}}, ($
obj, Motion, VELOCITY DIRECTION$)], [0], \theta_0)$
$less\_eq([v_{\text{large}}, (\text{sub, Motion, VELOCITY VALUE})], [0])$
($v_{\text{large}}$ defined by user)
$less\_eq([v_{\text{large}}, (\text{obj, Motion, VELOCITY VALUE})], [0])$
($v_{\text{large}}$ defined by user) |
| | from | $eq([\mathbf{p}^{\text{gt}}, (\text{obj, Motion, INITIAL POSITION})], [0])$
Here, $p_i^{\text{gt}} = p_i^{\text{sub}} \pm (s_i^{\text{sub}} + s_i^{\text{obj}} + C) * 0.5$ for $i \in [0, 2]$ |

Table 4: **Constraints in each multiple-object dynamic model.** These dynamic models are applied to at least two objects, one representing the subjective and the other representing the objective ("sub" and "obj" in the table). from and to relation will include another objective object, named "obj-extra" in the table. $C \geq 0$ represents random noise, and $\pm$ means +/- are chosen randomly in practice.

| DM | Type | Constraint |
|---|---|---|
| FLY | Basic | $similar\_dir([\mathbf{v}_{dir}^{\text{gt}}, (\text{sub, Motion, VELOCITY DIRECTION})], [0], \theta_0)$, $\quad \mathbf{v}_{dir}^{\text{gt}} = [C_0, 0, C_1]$ |
| | | $less\_eq([v_{\text{large}}, (\text{sub, Motion, VELOCITY VALUE})], [0])$ $\quad (v_{\text{large}} \text{ defined by user})$ |
| | | $less\_eq([\mathbf{p}^{\text{gt}}, (\text{sub, Motion, INITIAL POSITION})], [0])$, $\quad \mathbf{p}^{\text{gt}} = [p_0^{\text{gt}}, p_1^{\text{gt}}, p_2^{\text{gt}}], p_1^{\text{gt}}$ is user-defined threshold $\quad p_0^{\text{gt}}$ and $p_2^{\text{gt}}$ are the global minimum position |
| | from | $eq([\mathbf{p}^{\text{gt}}, (\text{obj, Motion, INITIAL POSITION})], [0])$ $\quad$ Here, $p_i^{\text{gt}} = p_i^{\text{sub}} \pm (s_i^{\text{sub}} + s_i^{\text{obj}} + C) * 0.5$ for $i \in [0, 2]$ |
| | to | $eq([\mathbf{p}^{\text{gt}}, (\text{obj, Motion, INITIAL POSITION})], [0])$ $\quad$ Here, $\mathbf{p}^{\text{gt}} = \mathbf{p}^{\text{sub}} + \mathbf{s}^{\text{sub}} + v_{value}^{\text{sub}} \cdot \mathbf{v}_{dir}^{\text{sub}} \cdot C$ |
| SLIDE | Basic | $similar\_dir([\mathbf{v}_{dir}^{\text{gt}}, (\text{sub, Motion, VELOCITY DIRECTION})], [0], \theta_0)$, $\quad \mathbf{v}_{dir}^{\text{gt}} = [C_0, 0, C_1]$ |
| | Basic (Not from) | $less\_eq([\mathbf{p}^{\text{gt-min}}, (\text{sub, Motion, INITIAL POSITION}), \mathbf{p}^{\text{gt-max}}], [0, 2])$ $\quad$ Here $\mathbf{p}^{\text{gt-*}} = [p_0^{\text{gt-*}}, p_1^{\text{gt-*}}, p_2^{\text{gt-*}}], p_1^{\text{gt-min}} = s_1^{\text{sub}}, p_1^{\text{gt-max}} = s_1^{\text{sub}} + C_{\text{small}}$ $\quad p_i^{\text{gt-*}}$ refers to global * position for $i = 0, 2$ and * refers to min/max |
| | from | $eq([\mathbf{p}^{\text{gt}}, (\text{sub, Motion, INITIAL POSITION})], [0])$, with $\mathbf{p}^{\text{gt}} = [p_0^{\text{gt}}, p_1^{\text{gt}}, p_2^{\text{gt}}]$ $\quad$ Here, $p_1^{\text{gt}} = p_1^{\text{obj}} + (s_1^{\text{sub}} + s_1^{\text{obj}} + C_{small}) * 0.5$ $\quad$ Random sample $p_i^{\text{gt}} \in [p_i^{\text{obj}} - s_i^{\text{obj}} * 0.5, p_i^{\text{obj}} + s_i^{\text{obj}} * 0.5]$ for $i = 0, 2$ |
| | to | $eq([\mathbf{p}^{\text{gt}}, (\text{obj, Motion, INITIAL POSITION})], [0])$ $\quad$ Here, $\mathbf{p}^{\text{gt}} = \mathbf{p}^{\text{sub}} + \mathbf{s}^{\text{sub}} + v_{value}^{\text{sub}} \cdot \mathbf{v}_{dir}^{\text{sub}} \cdot C$ |

Table 5: **Example of other possible dynamic models.** One can easily define and implement additional dynamic models with in our codebase.

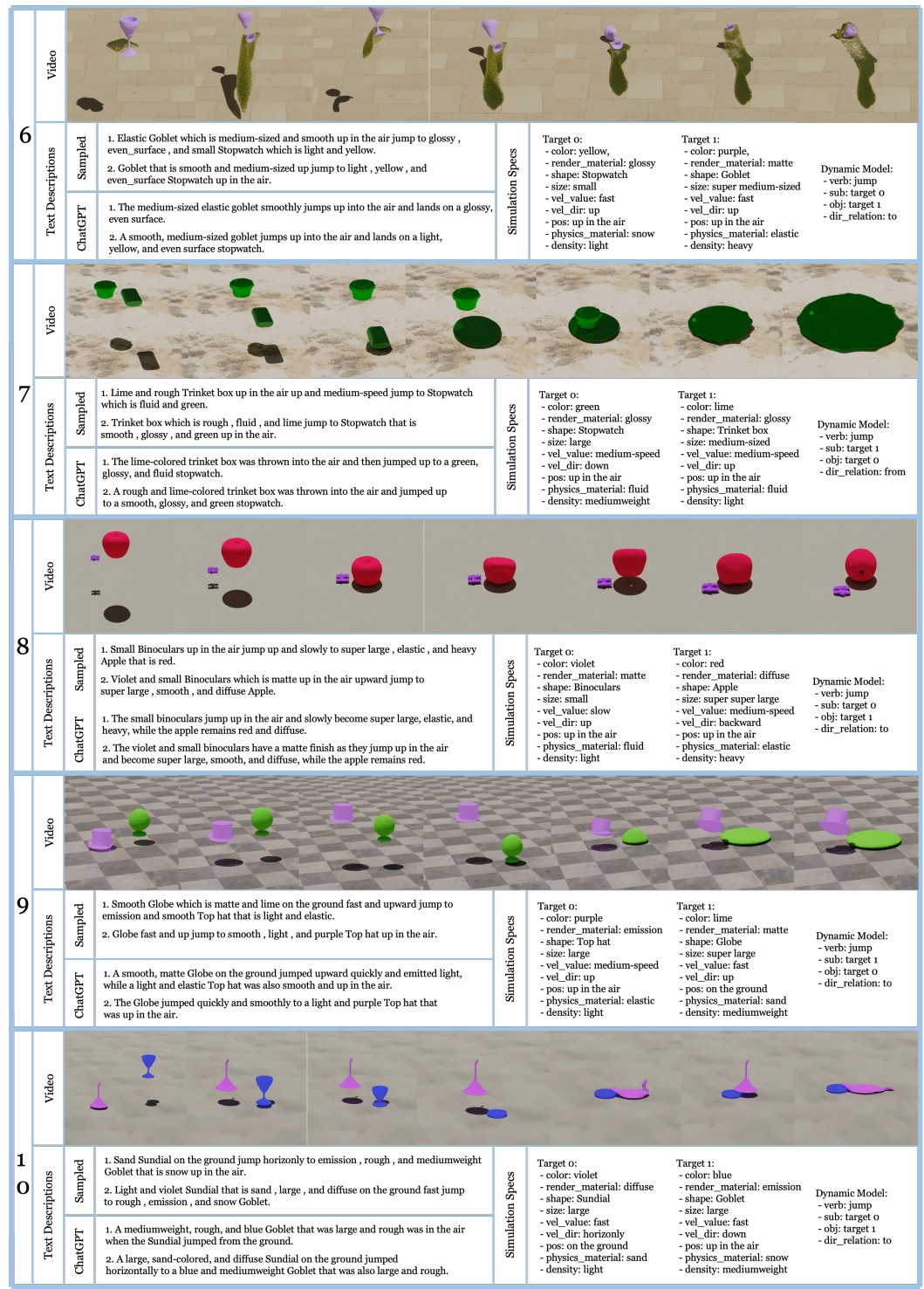

Figure 2: Data samples of **DROP** dynamics.

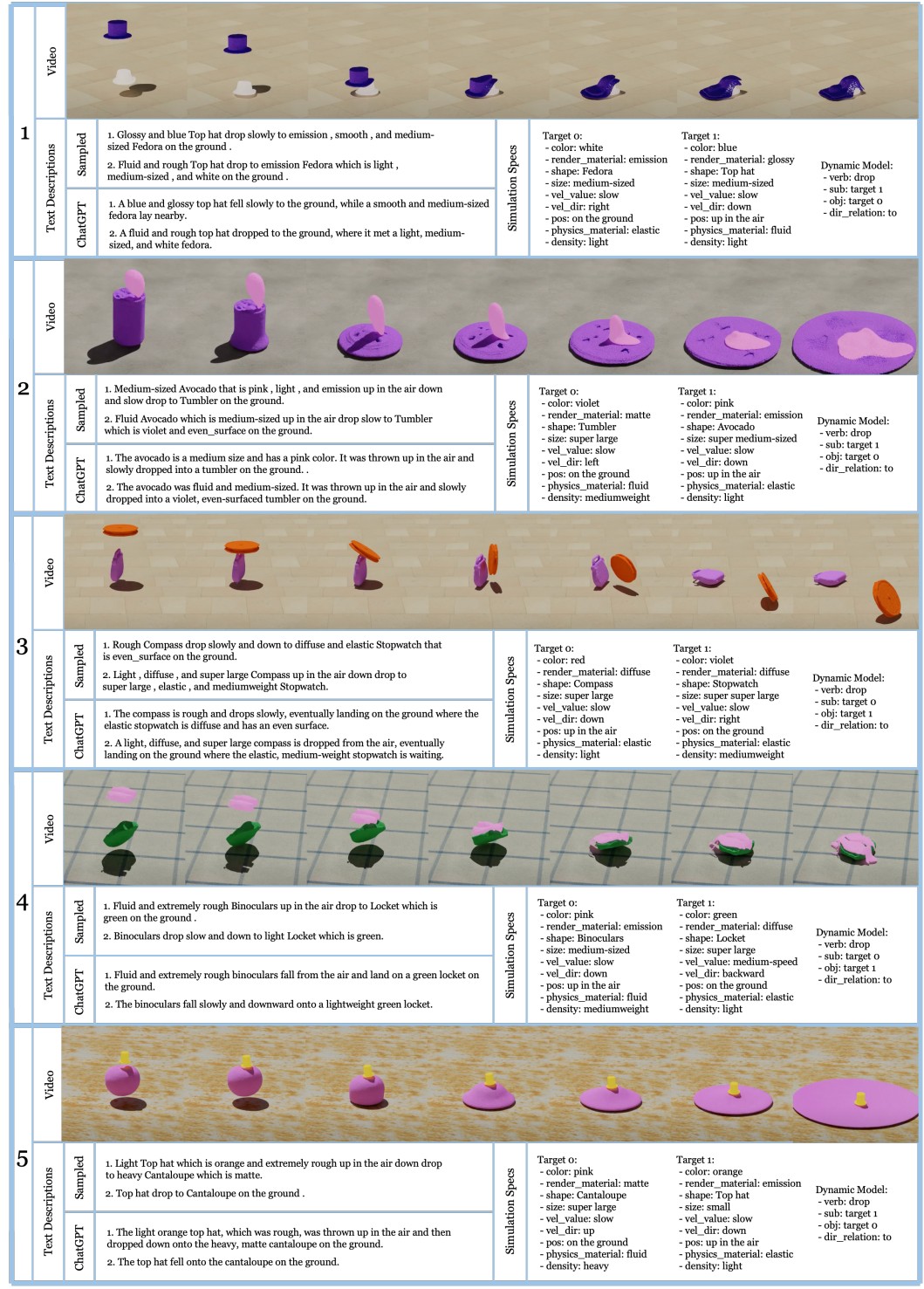

Figure 3: Data samples of **JUMP** dynamics.

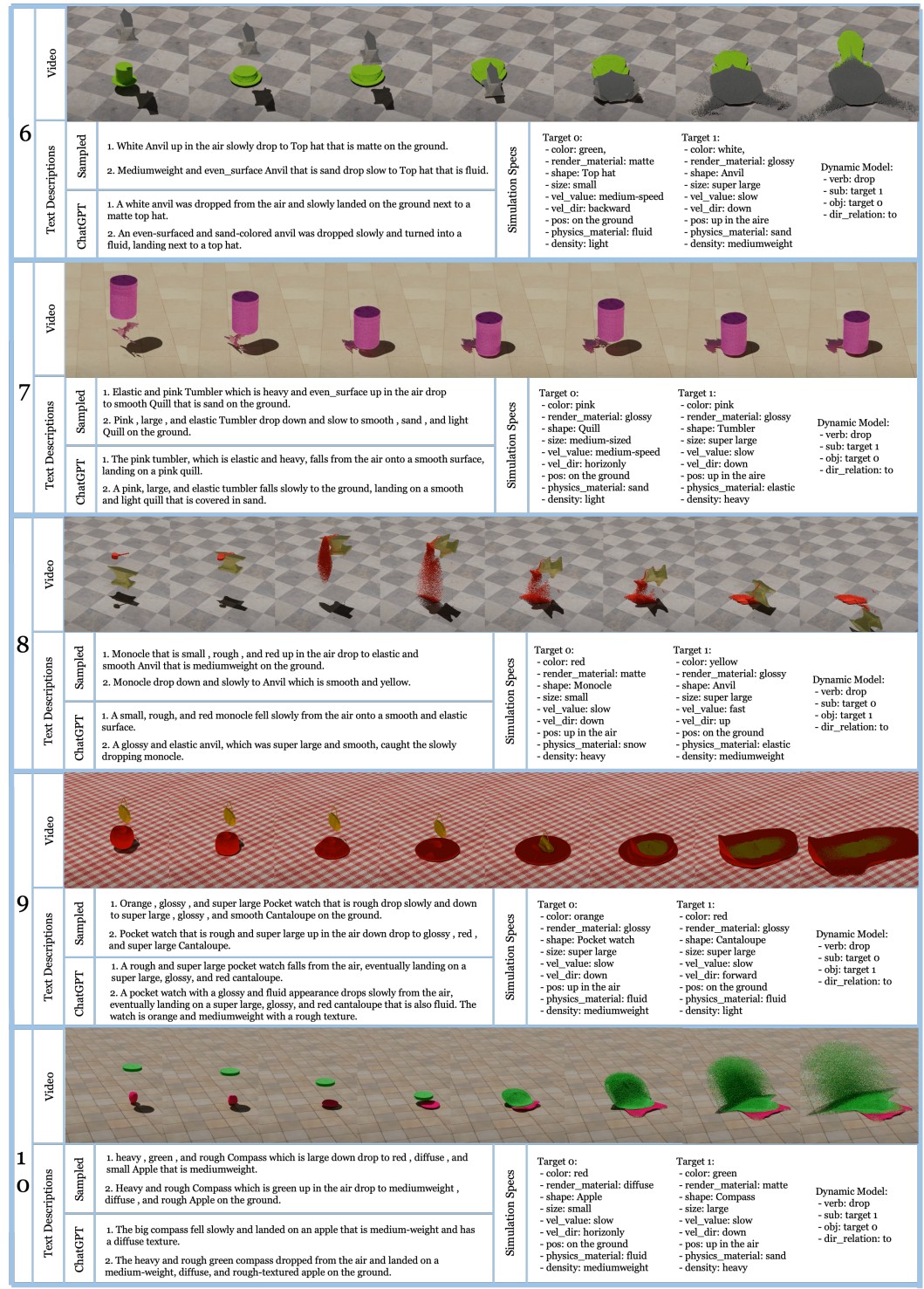

Figure 4: Data samples of **JUMP** dynamics.