# OpenReview forum: "TPA-Gen: A Multi-modal Data Generative Method for Text and Physics-based Animation"
_ICLR.cc/2024/Conference — ICLR 2024 Conference Withdrawn Submission_

### Official Review · Reviewer_dZiU · 2023-10-29

**Soundness:** 2 fair
**Presentation:** 3 good
**Contribution:** 2 fair
**Rating:** 1
**Confidence:** 3

**Summary:**

This paper presents an automated pipeline for text/simulation data generation. This includes randomly sampling scene grammars, object/motion attributes, rendering, as well as GPT models for text rephrase. The framework is able to generate high-quality 3D animations from a physical world, together with descriptive textual caption. This dataset aims for bridge the gap between text and vision in physical world.

**Strengths:**

1. First of all, this work contributes to an important problem in the vision community. Large text-video collection is difficult in practice, and usually these videos contain messy information, which further poses challenges for analysis in physical plausibility. The automated pipeline in this work can provide a much clearer data collection with high-quality videos, simulations, and texts.
2. I think the authors have properly leveraged existing tools and resources to develope such an pipeline.
3. The paper is well-written and easy to follow.

**Weaknesses:**

1. It is very weird to see the author list and institute in the supplementary file. Aren't the reviewing process of ICLR double-blind?
2. The proposed method is more like an engineering workflow, instead of "representation learning". The contribution primarily lies on a way to generate dataset, which however finally didn't yield a usable dataset, but a sample set. Even for the sample dataset, I also checked the provided link for possible video demonstration, but there is only a set of .ply files. Therefore, bases on current version, I would say many things seem like hypothetic or prototype, which limits the novelty and significancy of this work.
3. I think it would better to present some potential tasks that can be done on the datasets. Some simple baselines are also welcome.

**Questions:**

I think the violation of double-blind reviewing is sufficient for me to reject this work.

**Details Of Ethics Concerns:**

The author information is revealed in the supplementary files.

---

### Official Review · Reviewer_dKyv · 2023-10-31

**Soundness:** 3 good
**Presentation:** 3 good
**Contribution:** 2 fair
**Rating:** 5
**Confidence:** 3

**Summary:**

This paper proposes an automated pipeline to generate text-to-video physics simulation data. The authors leverage both solid and particle-based physics simulators. They conceive a grammar of scenes and physical events that enable sampling of random interactions of one or several objects with simple background environments, such as collisions between soft and rigid objects. They then sample a large dataset of physical interactions. Using the grammar, they further extract language descriptions of each video, that are refined using an LLM.

**Strengths:**

- The paper is well-motivated by current progress in vision-language models.
- A well-crafted physics dataset is also useful - there is a lack of high-quality physics simulations to learn from.
- Exposition of the paper is good, laying out the structure of scene & event grammar as well as the way the sentences are auto-generated from the sampled sequences.
- Related work is solid.
- I like that the authors already render out multi-view datasets, simplifying the 3D computer vision use case.

**Weaknesses:**

- My main concern is that (1) the events that are sampled are not realistic enough and (2) the sentence descriptions are similarly not particularly meaningful. For instance, in figure 4, we have "mediumweight anime". I have never heard anyone refer to a toy as an "anime". Further, the description says "super large" - the toy does not seem particularly large. Then, the verb is "is striking to" - that is not a reasonable verb. And finally, "a matte and snow vinyl on the ground" - that doesn't make much sense either, I don't know what a vinyl is (it's usually used to refer to either a material or vinyl records, but the video shows neither). In the next caption, we have  a "fluid, mediumweight and jolting treasure chest that is super large". Again, it doesn't seem particularly large. It is literally not calendered when the sequence begins, and is only calendered as the sequence continues. It is also not dropped from the container, rather, both of them are dropped from some height. The same is true for the captions in Fig. 5, which are also not particularly close to any caption that a human would have written.
- Conditioned on the low quality of the captions, I am unsure what purpose this dataset could serve in its current form. I believe the distribution of captions is too far removed from the distribution of captions that humans would use to describe the contents of these videos. Hence, what would one hope to learn from this dataset? I believe one would drastically overfit on this particular dataset, and I would expect very little transfer learning that could happen to any kind of real-world scene.
- This comes down to another weakness - evaluation. The typical way of evaluating a dataset like this one would be via a user study. In the present case, I would have expected that the authors collect human annotations for a part of the dataset, and then have an A/B study where human participants pick the captions according to a set of criteria, such as "level of detail". In the present case, I am unconvinced of the quality of the text captions.
- I think the present paper would be more impactful if the authors produce some kind of model that uniquely benefits from the present dataset, and maybe establish a proper benchmark. What kind of models do the authors think will be trained on this dataset? Can they maybe train a model themselves (even if it is a very basic model) and report a quantitative metric that other users can benchmark with? For instance, while the ShapeNet paper did not introduce a first model, they *did* introduce a quantitative benchmark.

**Questions:**

- What could be quantitative benchmarks that leverage your dataset?
- What would be the simplest model that you believe could benefit from your current dataset? Could you train such a model yourself as a baseline?
- How do you expect this dataset to be used?
- Why do you believe the current quality of captions to be sufficient?
- Did you consider collecting human annotation for this dataset? Why / why not?

---

### Official Review · Reviewer_ZNsp · 2023-11-06

**Soundness:** 1 poor
**Presentation:** 2 fair
**Contribution:** 2 fair
**Rating:** 3
**Confidence:** 3

**Summary:**

The paper proposes a method for automatically generating text-physics simulation video pairs that can be used for training text-to-simulation models.
At the core of the method is the design of set of attributed stochastic grammar that describes the 3D scene. A 3-step sample stragegy is designed so that different scenes can be sampled while satisfying all the constraints. The scenes are then simulated and rendered. The corresponding text descriptions can also be sampled using rules, and subsequently rewritten by ChatGPT. The 3D objects used for constructing the scenes are obtained from both existing datasets and Shap-E, a text-to-3D model.
Although no quantitative evaluation is provided, the paper compared the generated data with the output of existing text-to-video models, hypothesizing that the data could improve the physics understanding of these models.

**Strengths:**

* The proposed 3D scene & text generation pipeline is quite general and it can potentially benefit other data generation tasks that require the generation of scene-text pairs.
* Introducing physics into text-to-3d or text-to-video model is an understudied problem. And it is of great value to study whether such a problem can be solved solely through synthesized data.

**Weaknesses:**

* The entire paper is based on the assumption that the data generated through the proposed pipeline can improve the physics understanding of visual-language multimodal models. However, there is no experiment that verifies this assumption.
* Section 5 is flawed and incomplete. Instead of doing a proper AB testing by training a model using the generated data and evaluate the physical understanding of the models with and without the proposed data, the paper chose an extremely unfair setting of comparing the synthesized data directly with the results from an existing text-to-video model.
* Without any quantitative evaluate, it is not obvious whether the generated data is actually useful for its designated purpose. In fact, it can even be counterproductive due to the lack of photorealism and potentially non-realistic scene setup.
* Generating 3D scenes using stochastic grammar is not a new technique. It has been extensively studied, for example, in "Configurable 3D Scene Synthesis and 2D Image Rendering with Per-Pixel Ground Truth using Stochastic Grammars, Jiang et al., IJCV 2018"

**Questions:**

I wonder if there could be other ways to better evaluate the new dataset, even without training a generative model?